# Ablation of Slc26a6 Mitigates Myocardial Ischemia/Reperfusion Injury

**DOI:** 10.3390/biomedicines13122874

**Published:** 2025-11-25

**Authors:** Phung N. Thai, Lu Ren, Daphne A. Diloretto, Pauline Trinh, Valeriy Timofeyev, Ning Zong, Richard Q. Ngo, Nipavan Chiamvimonvat, Xiao-Dong Zhang

**Affiliations:** 1Department of Internal Medicine, Division of Cardiovascular Medicine, School of Medicine, University of California, Davis, Davis, CA 95616, USA; pnthai@ucdavis.edu (P.N.T.); luren@stanford.edu (L.R.); ddiloretto@mednet.ucla.edu (D.A.D.); pqntrinh@ucdavis.edu (P.T.); vytimofeyev@health.ucdavis.edu (V.T.); nzong@health.ucdavis.edu (N.Z.); richngo115@gmail.com (R.Q.N.); nchiamvimonvat@arizona.edu (N.C.); 2Department of Basic Medical Sciences, University of Arizona College of Medicine-Phoenix, Phoenix, AZ 85004, USA

**Keywords:** solute carrier, Slc26a6, ischemia, reperfusion, pH

## Abstract

**Background/Objectives**: Ischemic heart disease remains a leading cause of morbidity and mortality worldwide, accompanied by a major decline in local myocardial pH. However, the mechanisms of pH regulation and the homeostasis of H^+^ neutralizing buffers, such as HCO_3_^−^, in cardiomyocytes remain incompletely understood. We identified a solute carrier, Slc26a6, in mouse and human hearts playing key roles in the regulation of cardiac pH, excitability, and contractility. Slc26a6 is an acid loader, so we hypothesized that ablation of Slc26a6 may protect the heart from ischemia/reperfusion (I/R) injury. **Methods**: The I/R model was generated using wild type (WT) and Slc26a6 knockout (*Slc26a6^−/−^*) mice. Multidisciplinary in vivo, in vitro, and ex vivo approaches were used, including echocardiography, electrophysiology, hemodynamic monitoring, fluorescence microscopy, histochemistry, and cellular Ca^2+^ transients, sarcoplasmic reticulum Ca^2+^ load, and sarcomere shortening were recorded. **Results**: Troponin I level was lower in *Slc26a6^−/−^* I/R mice. *Slc26a6^−/−^* mice showed better systolic and diastolic function, reduced collagen deposition, and reduced infarct size compared to that of WT mice. Cellular experiments in measurement of sarcomere shortening, Ca^2+^ transients, and sarcoplasmic reticulum Ca^2+^ load in cardiomyocytes from the infarct zone supported the in vivo findings, demonstrating better single cell function in *Slc26a6^−/−^* compared to WT mice. Ex vivo pH_i_ measurement showed elevated pH_i_ in *Slc26a6^−/−^* mouse heart. **Conclusions**: Ablation of Slc26a6 protects the heart from I/R injury, suggesting the importance of Cl^−^/HCO_3_^−^ exchange in cardiac pH regulation and I/R injury. The elevated pH_i_ in *Slc26a6^−/−^* mouse heart may counterbalance the effects of the myocardium acidosis resulting from ischemia.

## 1. Introduction

Ischemic heart disease represents the most common type of heart disease, that can lead to heart failure, lethal ventricular arrhythmias, and sudden cardiac death. Cardiac ischemia induces a major decline in local myocardial pH resulting in depressed myocardial contractility, disturbances in cellular Ca^2+^ signaling, and cardiac arrhythmia [1,2,3,4,5]. However, the mechanisms of pH regulation in cardiomyocytes remain incompletely understood. The critical knowledge gaps stem from the fact that the proton neutralizing buffers such as HCO_3_^−^ have not been well studied. Several H^+^-equivalent transporters have been reported to contribute to cardiac pH regulation including Na^+^/H^+^ exchangers (NHEs), Na^+^-HCO_3_^−^ cotransporters (NBCs), anion exchangers or Cl^−^/HCO_3_^−^ exchangers (AEs or CBE), Cl^−^/OH^−^ exchangers (CHEs), and monocarboxylate transporters (MCTs) [1,2]. Ischemia leads to local myocardial acidosis due to increased anaerobic metabolism coupled with decreased vascular acid removal, accompanied by a downregulation of NHE and possible NBC activity [1,6]. Therefore, the intracellular pH (pH_i_) is significantly reduced during ischemia. Reperfusion enhances the acid extrusion through MCTs, NHEs, and NBCs [1,7,8].

NHEs and NBCs are Na^+^-dependent H^+^ extruders in the heart. In contrast, H^+^-equivalent transporters such as AEs and CHEs are chloride-coupled anion transporters in the heart associated with Cl^−^ and H^+^ loading into the cardiomyocytes. However, the molecular identity and physiological significance of these transporters in the heart are not well understood. Three AE family members AE1, AE2, and AE3 encoded by solute carrier *SLC4A* genes have been identified in the heart, with relative higher expression of AE3 [9,10,11,12]. AE1, AE2, and AE3 are electroneutral Cl^−^/HCO_3_^−^ exchangers [13]. AE3 was reported to be involved in the myocardial pH_i_ recovery from cellular alkalization [14], and ablation of AE3 showed phenotypical normal cardiac function and had no effect on the ischemia/reperfusion injury [15]. Further studies showed that loss of the AE3 in a hypertrophic cardiomyopathy model causes rapid decompensation and heart failure [16], and AE3 is required for cardiac protection by sasanquasaponin against ischemia/reperfusion (I/R) injury [17]. In addition, AE3 was recently proposed to be responsible for active transport-mediated disposal of CO_2_ in the heart [18]. In a zebrafish model, AE3 knockdown caused increased cardiac pH_i_, short QT syndrome, and reduced systolic duration [19].

Paradoxically, a recent study found that AE1, AE2, and AE3 are not the major Cl^−^/HCO_3_^−^ exchangers in the heart; instead, a new solute carrier, Slc26a6, is the predominant Cl^−^/HCO_3_^−^ and Cl^−^/OH^−^ exchanger with nearly 100-fold higher expression level than AE1, AE2, and AE3 in the heart [20]. To this end, we decided to investigate and have cloned several isoforms of mouse and human cardiac Slc26a6. We demonstrated that Slc26a6 is highly expressed in atrial and ventricular myocytes. Importantly, we showed that Slc26a6 mediates electrogenic Cl^−^/HCO_3_^−^ exchange activities in cardiomyocytes suggesting the potential role of Slc26a6 not only in the regulation of cardiac Cl^−^ homeostasis, but also in cardiac excitability and pH_i_ [21]. We tested the triple roles of Slc26a6 by taking advantage of the null deletion of *Slc26a6* and demonstrated that the null deletion results in action potential shortening, elevated pH_i_, fragmented QRS complexes, slower heart rate, and reduced cardiac function compared to WT littermates [22]. Therefore, Slc26a6 may represent a novel and predominant Cl^−^/HCO_3_^−^ exchanger in the heart with significant functional impact on cardiac function.

An early study found that intracellular Cl^−^ activities increase in ventricular muscles during simulated ischemia [23]. Inhibition of Cl^−^/HCO_3_^−^ exchange activities has been suggested to be a possible cardioprotective strategy against I/R injury [24]. In an I/R-induced ventricular fibrillation model, chloride substitution was suggested to be an effective antiarrhythmic approach [25,26]. Cl^−^ substitution has been shown to produce protective effects against I/R injury, and an anion exchange inhibitor, 4-acetamido-4′-isothiocyanato-stilbene-2,2′-disulfonic acid (SITS) was shown to protect the heart from I/R injury [24,27]. A recent study found that blockade of transmembrane Cl^−^ flux mitigates I/R-induced cardiac injury [28]. Slc26a6 represents the key Cl^−^/HCO_3_^−^ exchanger in the heart critical for cardiac Cl^−^ homeostasis and pH_i_ regulation. Here, we hypothesize that ablation of *Slc26a6* will result in a significant reduction of acid loading into cardiomyocytes, which may have protective effects on I/R injury. At the translational level, Slc26a6 may represent a novel therapeutic target for cardiac protection during I/R.

## 2. Materials and Methods

All animal care and procedures were performed in accordance with the protocols approved by the Institutional Animal Care and Use Committee of the University of California, Davis. Animal use was in accordance with the National Institutes of Health guidelines. We used 129S6/SvEv wild type (WT) and *Slc26a6*^−/−^ mice previously generated and reported [29]. The validation of the absence of Slc26a6 in the heart of the global *Slc26a6*^−/−^ mouse was reported in our previous publication supporting the deletion of Slc26a6 in the heart in this mouse model [22]. We used the same animal model in this study. For further validating the global *Slc26a6*^−/−^ mouse model used in this study, we provided the genotyping data as shown in Appendix A. All experiments described in the study were conducted in a blinded fashion. Specifically, the investigators who performed the physiological recordings, cell/tissue dissection, and staining had no knowledge of the genotypes of the animals. Chemicals used in this study were products of Sigma-Aldrich (St. Louis, MO, USA), unless specifically indicated.

### 2.1. I/R Mouse Model

Mouse I/R model was generated as we previously described [30]. Briefly, mice were injected intraperitoneally with ketamine 50–80 mg/kg and xylazine 5 mg/kg to achieve anesthesia during surgery. Mice were intubated and mechanically ventilated with isoflurane anesthetic (2%) and supplemental oxygen at a respiratory rate of 100–200 breaths/min and tidal volume of 0.13–0.2 mL. Surface ECG recording was recorded before and during the I/R surgery. An 8 mm cut was made 2 mm left of the sternal border in the 4th intercostal space. A retractor was used to gently widen the incision. The pericardium was clamped and fixed by the retractor. The left anterior descending (LAD) coronary artery was ligated 1–2 mm below the tip of the left auricle in its normal position, which induced approximately 40–50% ischemia of the LV. Occlusion was confirmed by the change of color of the anterior wall of the LV and ST elevation in ECG. The ligation was left for a period of 45 min after which the occlusion was removed. The retractor was then taken out, a temporary chest tube was inserted to re-establish the negative plural pressure until the spontaneous breathing recovered, and the chest cut was sutured. We performed the same operative procedures in sham mice except for LAD ligation. Additional postoperative treatment included supplemental buprenorphine twice daily as an analgesic for two days and a warm environment. Mice were monitored daily until the wound was healed.

### 2.2. Evaluation of Cardiac Function by Echocardiography

Echocardiograms using M-mode and two-dimensional (2D) measurements to assess systolic function were performed in conscious animals using Vevo 2100 (FUJIFILM VisualSonics, Toronto, ON, Canada), as we previously described [21,22]. Briefly, we performed at least two sperate scans to measure six cardiac cycles with papillary muscles as the position references. Left ventricle (LV) end diastolic and end systolic dimensions were measured for calculation of fractional shortening (FS) and ejection fraction (EF) defined as follows: FS% = (EDD − ESD)/EDD × 100, where EDD and ESD are LV end diastolic and end systolic dimension, respectively; EF% = (EDV − ESV)/EDV × 100, where EDV and ESV are LV end diastolic and end systolic volume, respectively.

### 2.3. Electrocardiographic Recordings

ECG recordings were performed using Bioamplifier (BMA 831, CWE, Inc., Ardmore, PA, USA), as we previously described [21,22]. Briefly, mice were placed on a temperature-controlled warming blanket at 37 °C. Four consecutive two-minute epochs of ECG data were recorded. Signals were low-pass filtered at 0.2 kHz and digitized using Digidata 1200 (Molecular Devices LLC., Sunnyvale, CA, USA). A total of 100 beats were analyzed from each animal in a blinded fashion.

### 2.4. Hemodynamic Monitoring

We performed hemodynamic surgery and measurement as we previously reported [22]. Briefly, Mice were anesthetized and maintained at 37 °C. We inserted the catheter with pressure and volume sensors into the LV through the carotid artery. The LV pressure and volume were recorded by a Millar Pressure–Volume System MPVS-300 (Millar, Inc., Houston, TX, USA), Power Lab, and Lab Chart 6.0 software (AD Instruments, Colorado Springs, CO, USA).

### 2.5. Cardiac Tissue Preparation and Cardiomyocyte Isolation

Cardiomyocyte isolation was performed as we reported [21,22]. We performed rapid heart excision, retrograde perfusion, and enzymatic dissociation.

### 2.6. Histological Analyses

Histological analyses of cardiac tissue were performed as we previously described [22]. Briefly, after washing out the blood, the mouse heart was embedded in paraffin. We then made precision cuts of heart slices (5 μm thick) along the short axis and stained the slices with Picrosirius red for collagen deposition assessment.

### 2.7. Measurement of Sarcomere Shortening and Ca^2+^ Transient (CaT)

The IonOptix sarcomere detection (IonOptix LLC., Westwood, MA, USA) and fast Fourier transform (FFT) method were used to measure sarcomere contraction as we previously reported [22]. The sarcomere contraction and CaT were measured simultaneously by the IonOptix cardiac contraction system. The sarcomere shortening was quantified by the percentage changes of sarcomere length during contraction relative to the diastolic length.

### 2.8. Ex Vivo pH_i_ Measurement by Confocal Imaging

The mouse was injected intraperitoneally with ketamine 50–80 mg/kg and xylazine 5 mg/kg to achieve anesthesia, and the heart was rapidly excised and placed in the cold Ca^2+^-free and HEPES-buffered Tyrode’s solution containing (in mM): 145 NaCl, 4 KCl, 1 MgCl_2_, 10 glucose, 0.33 NaH_2_PO_4_, 10 HEPES, and pH 7.4 by NaOH. After trimming the connected tissues, the aorta was cannulated, and retrograde perfusion was performed to clear the blood followed by perfusion for 30 min using 10 mL Ca^2+^-free and HEPES-buffered Tyrode’s solution containing 10 µM SNARF-1 at room temperature (22–24 °C) by circulating the solution using an electrical pump. The heart was subsequently perfused and washed twice by 10 mL Ca^2+^-free Tyrode’s solution each time. The heart was then transferred to a chamber with a cover glass bottom (#1.5 cover glass), mounted on the confocal microscope (LSM 700, Carl Zeiss, Oberkochen, Germany) stage, and perfused by HCO_3_^−^-buffered Tyrode’s solution containing the following (in mM): 120 NaCl, 24 NaHCO_3_, 4 KCl, 1 MgCl_2_, 2 CaCl_2_, 10 glucose, 0.33 NaH_2_PO_4_, gassed by 5% CO_2_ and 95% O_2_. Confocal imaging was performed by focusing on the epicardium of the heart. For pH_i_ measurement in the epicardial cells, the cells were excited by 555 nm laser and the emission images were acquired simultaneously at 580 and 640 nm using two band pass emission filters (585 ± 10 nm, and 630 ± 15 nm). The heart was perfused for 10 min by HCO_3_^−^-buffered Tyrode’s solution gassed by 5% CO_2_ and 95% O_2_, and the emission images were acquired for pH_i_ quantification. The SNARF emission ratio (F580/F640) was converted to a pH_i_ value using standard calibration as we reported before [22,31]. The calibration was performed by measuring the SNARF emission ratios (F580/F640) when cardiomyocytes were perfused by calibration solutions with five different pH values. The calibration solutions contained the following (in mM): 140 KCl, 1 MgCl_2_, 20 HEPES (or MES at pH 5.5), with pH 5.5, 6.5, 7.5, 8.5. Then, 10 µM nigericin (a K^+^/H^+^ antiporter ionophore) was added to the calibration solution before use.

### 2.9. Data Analysis and Statistics

The sample size of 5 was estimated based on the detection of at least 15% differences with alpha = 0.05 for a two-tailed test with the power > 0.95, with the standard deviation of the differences assumed to be 5% (SigmaStat, Grafiti LLC, Palo Alto, CA, USA). All data were included. Normal distribution was tested for choosing the appropriate statistical methods. For three or more groups, one-way ANOVA combined with Tukey’s post hoc analyses was used except where specified in the figure legends. For comparisons between two groups with equal sample numbers, two-sample *t*-test was used; for comparisons between unequal sample numbers in two groups, Welch’s *t*-test was used. If the data did not follow a normal distribution, non-parametric paired Wilcoxon signed-rank test was used. Statistical significance was defined as *p* < 0.05. Multiple software tools were used for statistical analyses including Origin Pro 2021 (OriginLab, Northampton MA, USA) and GraphPad Prism 9 (GraphPad Software, San Diego, CA, USA). Data are presented as mean ± S.E.M.

## 3. Results

### 3.1. I/R Mouse Model and Ischemia Monitoring During I/R Surgery

To test the effects of I/R on cardiac function and structural remodeling, we generated I/R models in both WT and *Slc26a6^−/−^* mice by performing left anterior descending coronary artery (LAD) ligation surgery in the heart. Figure 1 shows the experimental design to test the short-term and long-term effects of I/R injury. The reperfusion time was set to be 24 h and four weeks in short-term and long-term experiments, respectively.

For short-term reperfusion, we tested the serum troponin I level to evaluate myocardiac injuries; for long-term reperfusion, we investigated the effect on structural remodeling, cardiac function changes, hemodynamics alterations, and single cell function. We performed ECG recordings during I/R surgery to monitor the ischemia progress and the correlated ECG alterations with the surgery effects to be sure of the success of the ligation. Figure 2A shows the ECG traces recorded during I/R surgery. We found significant ST segment elevation in both WT and *Slc26a6^−/−^* I/R mice during LAD ligation surgery, supporting the success in making the I/R models in both WT and *Slc26a6^−/−^* mice. The ST elevation was more significant in WT mice during the I/R surgery as shown in Figure 2B.

### 3.2. Less Structural Remodeling in Slc26a6^−/−^ I/R Mice

To test the potential effects of I/R in WT and *Slc26a6^−/−^* mice, we compared the morphology of the heart between sham and I/R in WT and *Slc26a6^−/−^* mice. We found significant hypertrophy post I/R in both WT and *Slc26a6 ^−/−^* mice as shown in Figure 3A,B, supported by the increased heart weight/body weight ratios in both groups. To quantify the fibrosis, we performed Picrosirius red staining on heart slices to assess collagen deposition in infracted areas as shown in Figure 3C. The collagen deposition was further quantified as shown in Figure 3D. The percentage of collagen deposition is significantly higher in the I/R heart slice compared to the sham group. Interestingly, the deposition percentage is significantly smaller in the *Slc26a6^−/−^* I/R heart compared to the WT I/R heart, suggesting reduced injuries and structural remodeling in *Slc26a6^−/−^* mice. Consistently, the troponin I level in the plasma of *Slc26a6^−/−^* mice is significantly lower than that of WT mice as shown in Figure 3E.

### 3.3. Slc26a6 Knockout Reserved Cardiac Function in I/R Mice

The structural remodeling after I/R may cause alterations in cardiac function. To quantify cardiac function, we used echocardiography to assess the systolic and diastolic function in sham and I/R mice, focusing on the comparisons between WT and *Slc26a6^−/−^* mice. The representative images of echocardiography for systolic function in sham and I/R mice are shown in Figure 4A. We did not find significant changes in heart rate after I/R (Figure 4B), but the diastolic and systolic left ventricle volumes were significantly increased after I/R in both WT and *Slc26a6^−/−^* mice (Figure 4D,E). The ejection fraction (EF) and fractional shortening (FS) were also significantly reduced (Figure 4F,G). Interestingly, the systolic left ventricle volume was smaller and EF and FS are larger in *Slc26a6^−/−^* I/R mice compared to WT I/R mice, supporting the notion that the impairment of cardiac function in *Slc26a6^−/−^* I/R mice is less than that of WT I/R mice and demonstrating the protective roles of Slc26a6 deletion in I/R injury. To further assess the cardiac function, we measured the diastolic cardiac function as well. Figure 4H shows the representative images of echocardiography for testing diastolic function in sham and I/R mice. Analyses of the diastolic function showed reduced E/A ratio in WT I/R mice instead of *Slc26a6^−/−^* I/R mice (Figure 4I), and the E/A ratio in *Slc26a6^−/−^* I/R mice was significantly higher than that of WT I/R mice. In addition, the isovolumic relaxation time (IVRT) was prolonged in WT I/R mice but did not change in *Slc26a6^−/−^* I/R mice (Figure 4J), further supporting the protective roles of *Slc26a6* deletion in I/R injury.

### 3.4. Improved Hemodynamics in Slc26a6^−/−^ I/R Mice

To further characterize the cardiac function alterations, we performed hemodynamic recordings to directly assess the left ventricle mechanical properties in sham, WT I/R, and *Slc26a6^−/−^* I/R mice in the long-term reperfusion model. Figure 5A–C show the original recording traces of left ventricle pressure, volume, and the developed pressure (dP/dt), respectively. The summarized data are shown on the right panels. It was clearly shown that the end-systolic pressure is significantly reduced in WT I/R mice but not in *Slc26a6^−/−^* I/R mice. The end-systolic pressure in *Slc26a6^−/−^* I/R mice is higher than that in WT I/R mice, even the end-systolic pressure in *Slc26a6^−/−^* mice is lower than that of WT mice. In addition, the dP/dt in WT I/R mice is significantly smaller compared to the sham group. However, the dP/dt in *Slc26a6^−/−^* I/R mice did not have significant changes compared to that of sham mice supporting the protective roles of ablation of Slc26a6 in left ventricle function in *Slc26a6^−/−^* I/R mice.

### 3.5. Improved Sarcomere Contractility in Slc26a6^−/−^ Cardiomyocytes from I/R Mice

The in vivo functional studies support the protective roles of *Slc26a6^−/−^* deletion in I/R injury. To further understand the cellular mechanisms under this finding, we isolated the cardiomyocytes from the remote and infarct zones of the left ventricle after long-term I/R challenges. We measured the sarcomere shortening in the cardiomyocytes from sham and I/R mice as shown in Figure 6A,B. The sarcomere shortening was quantified and compared in Figure 6C between WT and *Slc26a6^−/−^* cardiomyocytes from sham, I/R remote zone, and I/R infarct zone, respectively. We found that the sarcomere shortening is significantly smaller in the cardiomyocytes from the infarct zone, with a trend of reduction in the cardiomyocytes from the remote zone but without statistical significance. In addition, the sarcomere shortening is less reduced in the cardiomyocytes from the infarct zone of *Slc26a6^−/−^* I/R mice compared to that of WT I/R mice, further supporting the protective roles of *Slc26a6^−/−^* deletion in I/R injury at single cellular level. To reveal the mechanisms, we measured the calcium transient (CaT) in cardiomyocytes from sham and I/R mice as shown in Figure 6D,E. The quantifications and comparisons of CaT between WT and *Slc26a6^−/−^* cardiomyocytes from sham, I/R remote zone, and I/R infarct zone are shown in Figure 6F. We found significant reduction of CaT in cardiomyocytes from remote and infarct zones in WT I/R mice, but not in *Slc26a6^−/−^* I/R mice. Indeed, the CaT amplitudes in cardiomyocytes from *Slc26a6^−/−^* I/R mice are significantly higher than that of WT I/R mice, consistent with the less reduced sarcomere shortening in *Slc26a6^−/−^* I/R mice, as shown in Figure 6C. To demonstrate the contractility and cell shortening kinetics of the single cardiomyocyte, representative sarcomere contraction time-course traces at different conditions are shown in Appendix A.

### 3.6. Elevated pH_i_ in Slc26a6^−/−^ Mouse Hearts

Slc26a6 is an acid loader in cardiomyocytes, and our previous study found that pH_i_ is elevated in cardiomyocytes isolated from *Slc26a6^−/−^* mice [22]. We reasoned that the elevated pH_i_ may play a role in the protection of I/R injury in *Slc26a6^−/−^* mice. To address this hypothesis, we developed an ex vivo confocal imaging technique to directly measure the epicardial pH_i_ in a perfused heart. Mouse hearts were perfused by HCO_3_^−^-buffered Tyrode’s solution, gassed by 5% CO_2_ and 95% O_2_. We used a ratiometric pH dye, SNARF-1, as pH_i_ indicator which was loaded into the heart by retrograde perfusion. During the measurement, the heart was perfused continuously. We found that the epicardial pH_i_ of the heart can be well quantified by confocal imaging shown in Figure 7A, and the pH_i_ in cardiomyocytes from *Slc26a6^−/−^* mouse heart is significantly higher than that from WT mice (Figure 7B), consistent with our cellular measurement reported before [22].

## 4. Discussion

Slc26a6 is the predominant Cl^−^/HCO_3_^−^ and Cl^−^/OH^−^ exchanger in the heart with significant roles in the regulation of cardiac action potentials, pH_i_, and cardiac function [21,22]. Slc26a6 serves as an acid loader in cardiomyocytes, responsible for transporting Cl^−^ into the cells and HCO_3_^−^ out of the cells, therefore, affecting the intracellular Cl^−^ homeostasis and pH_i_. Ischemia is characterized by a large fall of pH_i_ and extracellular pH (pH_o_) due to the lack of vascular perfusion with a significant increase in anaerobic metabolism and local partial pressure of CO_2_ [1,32,33,34]. Reperfusion normalizes the pH_o_ followed by a recovery of pH_i_ by activating NHE and NBC, resulting in increased intracellular Na^+^ and subsequent Ca^2+^ overload through the action of NCX.

Ablation of *Slc26a6* significantly elevated the pH_i_, and may play critical roles during I/R. Indeed, we found that *Slc26a6^−/−^* I/R mice demonstrated reduced structural remodeling, less impaired cardiac function, improved sarcomere shortening, and increased CaT in the in vivo and in vitro studies, compared with the WT I/R mice. Our ex vivo confocal imaging experiments showed an elevated pH_i_ in *Slc26a6^−/−^* mouse hearts supporting the important roles of pH_i_ regulation in the protection of I/R injury. Therefore, targeting pH_i_ regulatory mechanisms and the H^+^-equivalent transport proteins in cardiomyocytes may represent an effective strategy in preventing cardiac I/R injury.

### 4.1. I/R Injury

I/R injury is an ubiquitous pathological condition occurring in multiple organs. It is also a concern in organ and tissue transplantation procedures. In the heart, acute ischemia caused by coronary artery occlusions represents one of the leading causes of morbidity and mortality, and is the most common cause of chronic heart failure (HF) worldwide [35]. Malignant ventricular arrhythmia after acute myocardial infarction is the major cause of sudden cardiac death [36]. Myocardial necrosis and the adverse remodeling of myocardium are the results of I/R injury leading to endothelial dysfunction, microvascular injury, abnormal Ca^2+^ handling in myocardium, and altered myocardial metabolism and endogenous protective mechanisms [37,38,39]. Inflammation, oxidative stress, Ca^2+^ overloading, and mitochondrial dysfunction are the important mechanisms underlying I/R injury [35,36,37,40]. Despite the identification of multiple mechanisms and therapeutic strategies, the accumulated knowledge to date is not sufficient for the effective treatment of I/R injury. There are still critical knowledge gaps in the mechanistic understanding of myocardial I/R injury. The abnormal Ca^2+^ handling and endothelial and mitochondrial dysfunction are highly associated with ion transport including Na^+^, Ca^2+^, and H^+^ [41]. However, the mechanistic roles of HCO_3_^−^ and Cl^−^ are largely unknown. HCO_3_^−^ and Cl^−^ transport through ion channels and transporters is highly associated with cellular Ca^2+^ signaling and H^+^ homeostasis in the heart; we thus aimed to test the mechanistic roles of Slc26a6, a predominant cardiac Cl^−^/HCO_3_^−^ and Cl^−^/OH^−^ exchanger, in I/R injury. The significance is in that Cl^−^/HCO_3_^−^ and Cl^−^/OH^−^ exchange is tightly coupled with cardiac pH regulation, which is a key regulatory component in ischemia [2,42].

### 4.2. Abnormal pH Regulation in I/R

Myocardial ischemia is accompanied by a major decline in myocardial pH, resulting in depressed myocardial contractility, distorted cellular Ca^2+^ signaling, and cardiac arrhythmia [1,2,3,4,5]. Local acidosis is a hallmark of ischemia, resulting from increased anaerobic metabolism, lack of vascular perfusion, and an increase in local partial pressure of CO_2_. The significance of pH regulation lies in that the homeostasis and transport of several important ions including Na^+^, Ca^2+^, K^+^, Cl^−^, and HCO_3_^−^ are highly dependent on pH_i_ and pH_o_ [1,6,43,44,45,46,47,48,49,50,51]. In addition, pH_i_ and pH_o_ regulate the function of contractile proteins and enzymatic activities of critical signaling pathways that are disrupted in I/R injury [32,35,36,38,40,52,53,54,55,56,57,58,59,60]. Therefore, dysregulation of pH_i_ and pH_o_ plays an important role in I/R injury. However, the molecular mechanisms of pH regulation in I/R are still not fully understood, and the specific molecular targets responsible for I/R injury have not been well studied and clearly identified. NHE and NBCs are Na^+^-dependent acid extruders responsible for pH_i_ and intracellular Na^+^ regulations, affecting the intracellular Ca^2+^ homeostasis. The functional role of NHE has been extensively studied in regards to I/R injury, and the inhibition of NHE in I/R animal models has shown protective effects against injury through reductions of intracellular Ca^2+^ overload [6,61,62,63,64,65,66,67,68,69,70]. However, clinical trials using NHE1 inhibitors as cardioprotective agents during I/R proved to be disappointing and without significant benefits, and they even presented certain risks of increased incidence of stroke when higher doses of some agents were used [1,61,64,71]. The role of NBCs has also been studied. Inhibition of NBCe was reported to reduce I/R injury [72,73,74,75,76], and ablation of NBCe1 caused reduced apoptosis following I/R injury [77]. Due to the complicated regulatory mechanisms and lack of specific inhibitors of NBC, the mechanistic roles of NBC in I/R injury need to be further studied. As we and others reported previously, Slc26a6 is the predominant Cl^−^/HCO_3_^−^ exchanger in the heart with significant roles in the regulation of cardiac excitability and pH_i_ [20,21,78]. In this study, we took advantage of the *Slc26a6*^−/−^ mouse model to investigate the specific roles of Slc26a6 in I/R injury. We found less injury in *Slc26a6*^−/−^ mice, supporting the importance of Slc26a6 in I/R injury protection. By establishing a new confocal microscopy technique to measure epicardial pH_i_ in an ex vivo heart model, we identified elevated pH_i_ in *Slc26a6*^−/−^ mouse heart. The elevated pH_i_ in *Slc26a6*^−/−^ mouse heart may counter-balance the acidification caused by ischemia to provide potential protection in I/R injury.

### 4.3. Role of Chloride Transporters in I/R Injury

In cardiomyocytes, one critical group of H^+^-equivalent transporters are chloride-coupled anion transporters, including AEs and CHEs. The AE family has three members, AE1, AE2, and AE3, with relative higher expression of AE3 in the heart [9,10,11,12]. AEs are electroneutral Cl^−^/HCO_3_^−^ exchangers [13]. AE3 was reported to mediate pH_i_ recovery from cellular alkalization [14], but ablation of AE3 did not affect cardiac function with no significant demonstrable roles in I/R injury [15]. AE3 knockdown in zebrafish increased cardiac pH_i_, shortened QT, and reduced systolic duration [19]. However, AE1, AE2, and AE3 are not the major Cl^−^/HCO_3_^−^ exchangers in the heart; instead, Slc26a6 is the predominant Cl^−^/HCO_3_^−^ and Cl^−^/OH^−^ exchanger [20]. Our previous studies demonstrated that Slc26a6 is highly expressed in atrial and ventricular myocytes and mediates electrogenic Cl^−^/HCO_3_^−^ exchange [21], while ablation of *Slc26a6* causes action potential shortening, elevated pH_i_, fragmented QRS complexes, slower heart rate, and reduced cardiac function [22]. Slc26a6 represents a predominant Cl^−^/HCO_3_^−^ exchanger in the heart with important roles in the regulation of cardiac excitability, pH_i_, and function. However, the mechanistic roles of Slc26a6 in cardiac I/R injury are unknown.

It was reported that intracellular Cl^−^ activities increased in ventricular muscles during simulated ischemia [23], suggesting the important roles of enhanced chloride transport into cardiomyocytes in ischemia. Interestingly, inhibition of Cl^−^/HCO_3_^−^ exchange was proposed to be a novel cardioprotective strategy against I/R injury [24]. Reduced intracellular Cl^−^ activities by Cl^−^ substitution produced protective effects against I/R injury, and inhibition of Cl^−^ transport by 4-acetamido-4′-isothiocyanato-stilbene-2,2′-disulfonic acid (SITS) showed protective roles in I/R injury [24,27]. Moreover, blockade of transmembrane Cl^−^ flux mitigates I/R injury via the inhibition of calpain activity [28].

Slc26a6 represents the predominant Cl^−^/HCO_3_^−^ and Cl^−^/OH^−^ exchanger in the heart responsible for cellular Cl^−^ homeostasis and pH_i_ regulation. Slc26a6 is an acid and chloride loader, facilitating inward chloride and H^+^ transport. Ischemia causes significant myocardium acidification with a large fall of pH_i_ and pH_o_, and the reperfusion removes the acid and activates NHE and NBC for pH_i_ recovery.

In the current study, we directly demonstrated that ablation of *Slc26a6* reduced acid loading and increased pH_i_ in cardiomyocytes [22] and in ex vivo hearts (Figure 7B), while the fall of cardiac pH_i_ in *Slc26a6^−/−^* mice during ischemia may be ameliorated compared to WT mice. As a result, the recovery of pH_i_ during reperfusion induces NHE activation to a lesser extent, and the disturbance to intracellular Na^+^ and Ca^2+^ is relatively smaller. Our data supported the important roles of elevated pH_i_ in *Slc26a6^−/−^* mice, and this may represent one of the mechanisms for the reduced I/R injury in *Slc26a6^−/−^* mice. In addition, shortened action potentials and reduced CaT in *Slc26a6^−/−^* cardiomyocytes may reduce Ca^2+^ influx and intracellular Ca^2+^ overload during reperfusion, which may protect the heart from injury as well.

Moreover, the reduced chloride loading in *Slc26a6^−/−^* I/R mice may represent another potential mechanism. Increased intracellular Cl^−^ activities in ischemia have been reported [23], and reduced intracellular Cl^−^ activities by inhibition of Cl^−^ transport or Cl^−^ substitution both have protective effects on I/R injury [24,27,28]. The underlying mechanisms may be related to the coupled Cl^−^/HCO_3_^−^ exchange, intracellular Ca^2+^ overload, attenuation of oxidative stress, inhibition of NF-kappaB activation [27], anion exchange stimulation through protein kinase C activation [24], and the inhibition of calpain activity [28]. Further mechanistic study is necessary for the in-depth understanding of the roles of Cl^−^ in the regulation of cellular function in I/R.

## 5. Conclusions

Slc26a6 plays critical roles in the regulation of cardiac pH_i_, excitability, and contractility. Due to its essential role in acid loading in cardiomyocytes, it is a potential target for correcting the abnormal pH_i_ in pathological conditions including I/R with dysregulations of pH_o_ and pH_i_. Our study demonstrated that ablation of Slc26a6 affords cardioprotective effects from I/R injury, highlighting the importance of Cl^−^ and pH regulation in I/R. Targeting Cl^−^/HCO_3_^−^ exchange mediated by Slc26a6 and other transporters may represent a new therapeutic strategy in the management of I/R injury.

## 6. Future Studies

Our study revealed that the elevated pH_i_ in *Slc26a6^−/−^* mouse heart may be a potential mechanism in mitigating I/R injury. However, more studies are needed to directly test the impact of pH modulation, including using the ex vivo I/R model for pH_i_ measurement, Langendorff perfusion strategy, and force and pressure measurement. Importantly, the downstream molecular mechanisms are still not well defined, and more efforts are needed to test the expression and function of other H^+^ equivalent transporters and ion channels, as well as pH-sensitive processes and mitochondrial function in the heart. Moreover, platelets play a crucial role in the occurrence and development of myocardial ischemia/reperfusion injury [79,80]. Some SLC gene family members have been identified in platelets [81], but whether Slc26a6 is expressed in platelets has not been addressed. Slc26a6 mediated Cl^−^/HCO_3_^−^ exchange may also be involved in the regulation of platelet function. Future studies will be performed to test whether Slc26a6 is expressed in platelets and how it regulates platelet function in I/R injury.

## Figures and Tables

**Figure 1 biomedicines-13-02874-f001:**
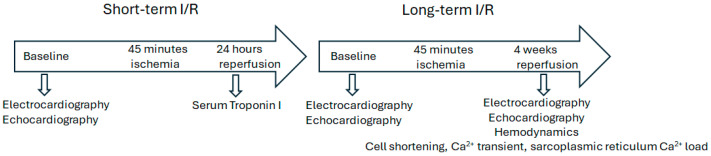
**I/R experimental design.** The mouse I/R model and experimental design are shown. Both WT and *Slc26a6^−/−^* male and female mice were used. Sham mice were generated by performing similar surgical incision but without left anterior descending coronary artery (LAD) ligation.

**Figure 2 biomedicines-13-02874-f002:**
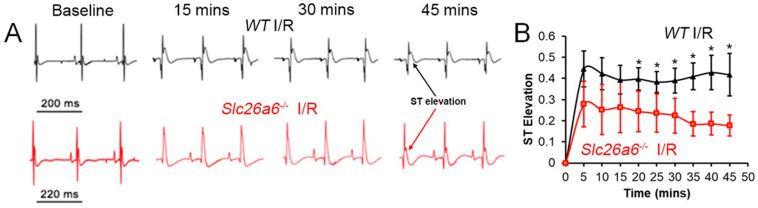
**ECG monitoring during I/R surgery**. (**A**) Representative traces of surface ECG during the LAD ligation surgery in WT and *Slc26a6^−/−^* mice. (**B**) Comparisons of ST segment elevation during LAD ligation. Biological replicates or mouse numbers: WT (12), *Slc26a6^−/−^* (12). (* *p* < 0.05 by two-sample *t*-test).

**Figure 3 biomedicines-13-02874-f003:**
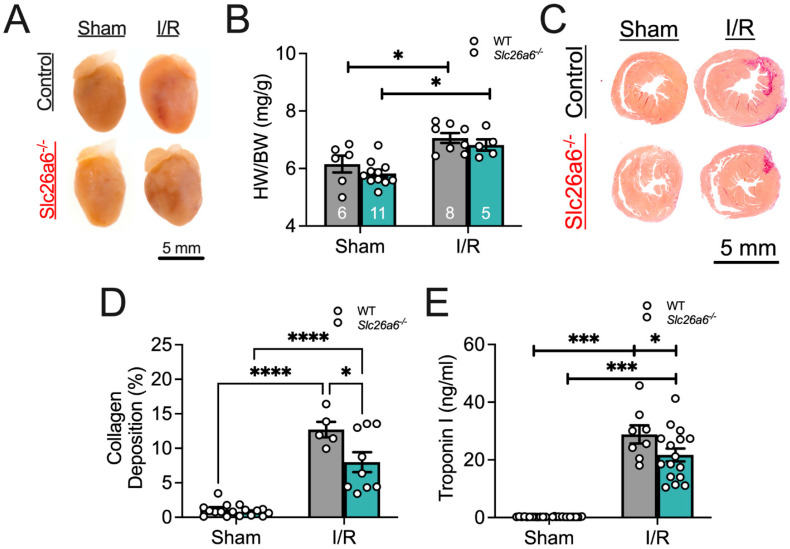
**Comparisons of cardiac I/R injury and structural remodeling**. (**A**) Heart images before and after I/R. (**B**) Comparisons of the ratios of heart weight to body weight (HW/BW) in WT and *Slc26a6^−/−^* mice from long-term I/R model. Biological replicates or mouse numbers: WT sham (6); *Slc26a6^−/−^* sham (11); WT I/R (8); *Slc26a6^−/−^* I/R (5). (* *p* < 0.05 by one-way ANOVA combined with Tukey–Kramer post hoc analyses). The data points in bar figures show the animal number for different experiments. Grey color bar: WT; cyan blue color bar: *Slc26a6^−/−^*. (**C**) Fibrosis analysis using Picrosirius red staining to assess the collagen deposition. (**D**) Summary data of cardiac collagen deposition in WT and *Slc26a6^−/−^* mouse from long-term I/R model. Biological replicates or mouse numbers: WT sham (9); *Slc26a6^−/−^* sham (8); WT I/R (5); *Slc26a6^−/−^* I/R (9). (* *p* < 0.05, **** *p* < 0.0001 by one-way ANOVA combined with Tukey–Kramer post hoc analyses). Grey color bar: WT; cyan blue color bar: *Slc26a6^−/−^*. (**E**) Comparisons of serum troponin I levels in WT and *Slc26a6^−/−^* mice from short-term I/R model. Biological replicates or mouse numbers: WT sham (9); *Slc26a6^−/−^* sham (8); WT I/R (8); *Slc26a6^−/−^* I/R (16). (* *p* < 0.05, *** *p* < 0.001 by one-way ANOVA combined with Tukey–Kramer post hoc analyses). Grey color bar: WT; cyan blue color bar: *Slc26a6^−/−^*.

**Figure 4 biomedicines-13-02874-f004:**
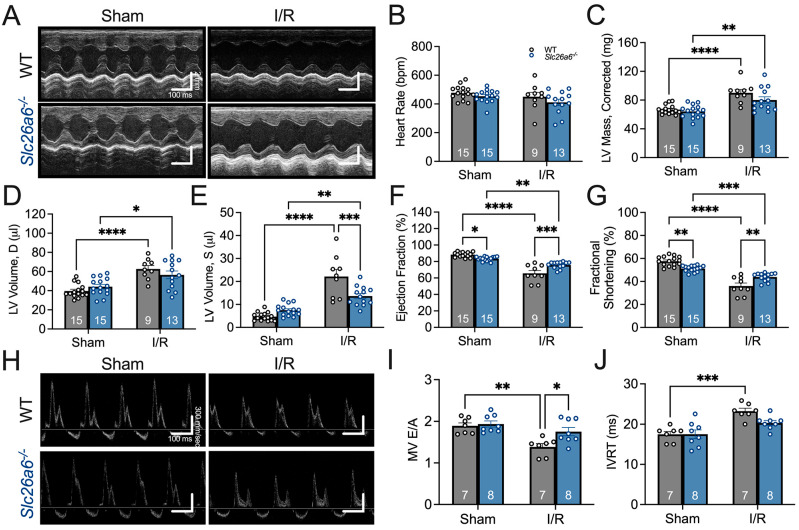
**Comparisons of cardiac function between WT and *Slc26a6^−/−^* I/R mice**. (**A**) Representative images of M-mode echocardiogram before and after I/R surgery in WT and *Slca6a6^−/−^* mice. (**B**) Comparisons of heart rates. Biological replicates or mouse numbers: WT sham (15); *Slc26a6^−/−^* sham (15); WT I/R (9); *Slc26a6^−/−^* I/R (13). (**C**) Comparisons of left ventricle (LV) mass. Biological replicates or mouse numbers: WT sham (15); *Slc26a6^−/−^* sham (15); WT I/R (9); *Slc26a6^−/−^* I/R (13). (** *p* < 0.01, **** *p* < 0.0001 by one-way ANOVA combined with Tukey’s post hoc analyses). (**D**) Comparisons of diastolic LV volumes. Biological replicates or mouse numbers: WT sham (15); *Slc26a6^−/−^* sham (15); WT I/R (9); *Slc26a6^−/−^* I/R (13). (* *p* < 0.05, **** *p* < 0.0001 by one-way ANOVA combined with Tukey’s post hoc analyses). The LV end-diastolic dimensions (in mm) are as follows: WT sham, 3.17 ± 0.08; WT I/R, 3.59 ± 0.14; Slc26a6^−/−^ sham, 3.31 ± 0.08; Slc26a6^−/−^ I/R, 3.61 ± 0.13. (**E**) Comparisons of systolic LV volume. Biological replicates or mouse numbers: WT sham (15); *Slc26a6^−/−^* sham (15); WT I/R (9); *Slc26a6^−/−^* I/R (13). (** *p* < 0.01, *** *p* < 0.001, **** *p* < 0.0001 by one-way ANOVA combined with Tukey’s post hoc analyses). The LV end-systolic dimensions are as follows: WT sham, 1.35 ± 0.08; WT I/R, 2.36 ± 0.13; Slc26a6^−/−^ sham, 1.57 ± 0.06; Slc26a6^−/−^ I/R, 2.04 ± 0.08. (**F**) Comparisons of ejection fractions. Biological replicates or mouse numbers: WT sham (15); *Slc26a6^−/−^* sham (15); WT I/R (9); *Slc26a6^−/−^* I/R (13). (* *p* < 0.05, ** *p* < 0.01, *** *p* < 0.001, **** *p* < 0.0001 by one-way ANOVA combined with Tukey’s post hoc analyses). (**G**) Comparisons of fractional shortening. Biological replicates or mouse numbers: WT sham (15); *Slc26a6^−/−^* sham (15); WT I/R (9); *Slc26a6^−/−^* I/R (13). (** *p* < 0.01, *** *p* < 0.001, **** *p* < 0.0001 by one-way ANOVA combined with Tukey’s post hoc analyses). (**H**) Echocardiogram images showing cardiac diastolic function. (**I**) Comparisons of E/A rations. Biological replicates or mouse numbers: WT sham (7); *Slc26a6^−/−^* sham (8); WT I/R (7); *Slc26a6^−/−^* I/R (8). (* *p* < 0.05, ** *p* < 0.01 by one-way ANOVA combined with Tukey’s post hoc analyses). (**J**) Comparisons of isovolumic relaxation time (IVRT). Biological replicates or mouse numbers: WT sham (7); *Slc26a6^−/−^* sham (8); WT I/R (7); *Slc26a6^−/−^* I/R (8). (*** *p* < 0.001 by one-way ANOVA combined with Tukey’s post hoc analyses). The data points in bar figures show the animal number for different experiments.

**Figure 5 biomedicines-13-02874-f005:**
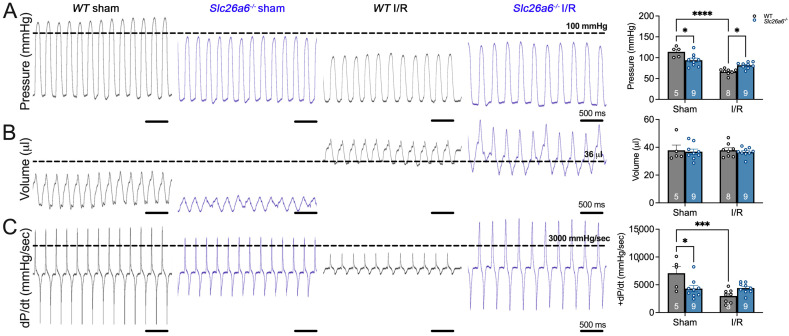
**Hemodynamic monitoring of the left ventricle function in WT and *Slc26a6^−/−^* I/R mice**. (**A**) Representative traces of left ventricular pressures (left). The comparison of the systolic left ventricular pressures is shown on the right. Biological replicates or mouse numbers: WT sham (5); *Slc26a6^−/−^* sham (9); WT I/R (8); *Slc26a6^−/−^* I/R (9). (* *p* < 0.05, **** *p* < 0.0001 by one-way ANOVA combined with Tukey’s post hoc analyses). (**B**) Representative traces of left ventricular volumes (left). The comparison of the diastolic left ventricular volumes is shown on the right. Biological replicates or mouse numbers: WT sham (5); *Slc26a6^−/−^* Sham (9); WT I/R (8); *Slc26a6^−/−^* I/R (9). (**C**) Representative traces of left ventricular pressure development (derivative of pressure with respect to time, dP/dt). The comparison of the pressure development is shown on the right. Biological replicates or mouse numbers: WT sham (5); *Slc26a6^−/−^* sham (9); WT I/R (8); *Slc26a6^−/−^* I/R (9). (* *p* < 0.05, *** *p* < 0.001 by one-way ANOVA combined with Tukey’s post hoc analyses). The data points in bar figures show the animal number for different experiments.

**Figure 6 biomedicines-13-02874-f006:**
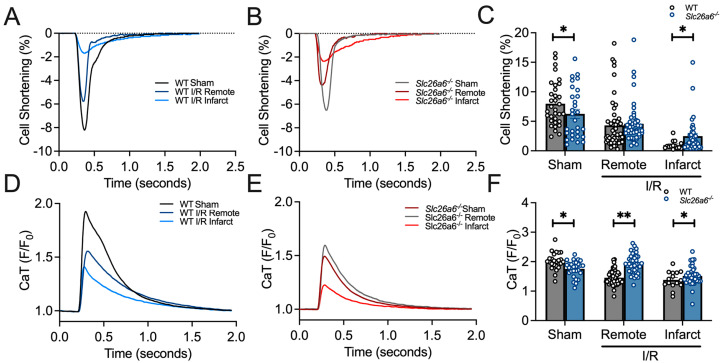
**Measurement and quantification of sarcomere contraction in WT and *Slc26a6^−/−^* I/R cardiomyocytes**. (**A**,**B**) Representative traces of sarcomere shortening quantified by single cardiomyocyte recordings from WT, *Slca6a6^−/−^* I/R and sham mice. (**C**) Summary data of sarcomere shortening in cardiomyocytes isolated from remote and infarct zones of the heart from WT, *Slc26a6^−/−^* I/R, and sham mice. Biological replicates including cell numbers and mouse numbers: WT sham (34 cells from 12 mice); *Slc26a6^−/−^* sham (29 cells from 12 mice); WT I/R remote zone (42 cells from 12 mice); *Slc26a6^−/−^* I/R remote zone (45 cells from 12 mice); WT I/R infarct zone (16 cells from 12 mice); *Slc26a6^−/−^* I/R infarct zone (30 cells from 12 mice). Comparisons were performed between WT and *Slc26a6^−/−^* cardiomyocytes (* *p* < 0.05 by Welch’s *t*-test). (**D**,**E**) Representative traces of CaT quantified by single cardiomyocyte recordings from WT, *Slc26a6^−/−^* I/R and sham mice. (**F**) Summary data of CaT peak amplitude in cardiomyocytes isolated from remote and infarct zones of the heart from WT, *Slc26a6^−/−^* I/R and sham mice. Biological replicates including cell numbers and mouse numbers: WT sham (26 cells from 12 mice); *Slc26a6^−/−^* sham (32 cells from 12 mice); WT I/R remote zone (30 cells from 12 mice); *Slc26a6^−/−^* I/R remote zone (40 cells from 12 mice); WT I/R infarct zone (16 cells from 12 mice); *Slc26a6^−/−^* I/R infarct zone (30 cells from 12 mice). Comparisons were performed between WT and *Slc26a6^−/−^* cardiomyocytes (* *p* < 0.05, ** *p* < 0.01 by Welch’s *t*-test).

**Figure 7 biomedicines-13-02874-f007:**
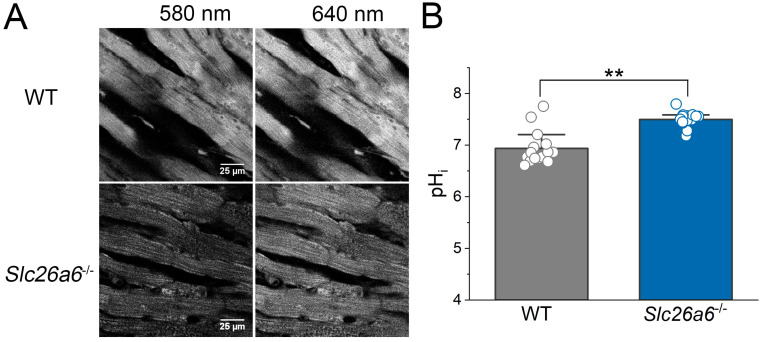
**Elevated pH_i_ in *Slc26a6^−/−^* mouse hearts revealed by ex vivo pH_i_ measurement using confocal imaging**. (**A**) Representative confocal images of epicardial cells loaded with SNARF-1 from WT and *Slc26a6*^−/−^ hearts. (**B**) Quantifications of epicardial pH_i_ of WT and *Slc26a6^−/−^* mice. Biological replicates including cell numbers and mouse numbers: WT (15 cells from 6 mice); *Slc26a6^−/−^* (15 cells from 3 mice). (** *p* < 0.01 by two-sample *t*-test).

## Data Availability

The data supporting the current study have not been deposited in a public repository because they have only been used for the publication of this manuscript, but are available from the corresponding author on request.

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
