# Peer review of "Ablation of Slc26a6 Mitigates Myocardial Ischemia/Reperfusion Injury"

_biomedicines, 2025, doi:10.3390/biomedicines13122874_

Round 1
Reviewer 1 Report
Comments and Suggestions for Authors
The study addresses a critical gap in understanding myocardial pH regulation during ischemia/reperfusion (I/R) injury, which contribute to cardiovascular research.
1, The functional status of platelets plays a crucial role in the occurrence and development of myocardial ischemia/reperfusion injury. The author utilized a conventional knockout mouse model. It is necessary to determine whether Slc26a6 is expressed in platelets, and if so, whether it influences platelet function. If Slc26a6 does affect platelet function, then whether Slc26a6 consequently impacts myocardial ischemia/reperfusion injury through modulating platelet function should be investigated. An appropriate discussion of this issue should be included in the discussion section of the article.
2, The study links Slc26a6 ablation to elevated pH and improved I/R outcomes, but it does not directly test whether pH modulation is the primary mechanism.
Reviewer 2 Report
Comments and Suggestions for Authors
This study examines the cardioprotective role of Slc26a6 deletion in a mouse ischemia/reperfusion (I/R) model. While the hypothesis that Slc26a6-mediated acid loading contributes to ischemic injury is novel and of interest, several aspects of the manuscript require clarification or improvement. Particularly, the functional data remain insufficiently substantiated, and key conclusions may be overly speculative based on the current evidence.
Major Points
1. Unclear Changes in Cardiac Dimensions (LVDd, LVDs)
-
Concern: The manuscript reports improvements in systolic and diastolic function (EF, FS) based on echocardiography. However, raw data for left ventricular end-diastolic and end-systolic diameters (LVDd, LVDs) are not shown.
-
Suggestion: Please include full quantification of LVDd and LVDs for all groups, as these are essential for validating EF/FS differences. From the representative images, the changes appear modest, raising concerns about whether the EF/FS values are truly different.
2. Overstatement in the Introduction
-
Sentence: “Ischemic heart disease is the most common type of heart disease.”
-
Concern: While ischemic heart disease is indeed prevalent, this opening statement may be overly generalized.
-
Suggestion: Consider rephrasing to something more nuanced and accurate, such as:
“Ischemic heart disease remains a leading cause of morbidity and mortality worldwide.”
3. Speculative Conclusions Without Mechanistic Data
-
Concern: While Slc26a6 ablation appears to improve cardiac function, the downstream molecular mechanisms are not well defined.
-
Suggestion:
-
Consider measuring expression or activity of acid/base transporters, pH-sensitive ion channels, or mitochondrial function to support the proposed model.
-
4. Sarcomere Shortening as a Functional Readout
-
Concern: Sarcomere shortening is presented as a key readout of preserved contractility, but there is no visual or supplementary representation (e.g., images, traces).
-
Suggestion:
-
Include representative sarcomere traces or time-course curves, or a supplementary figure showing shortening kinetics.
-
Consider validating contractility findings using complementary methods such as cellular fractional shortening, force measurements, or Langendorff perfusion pressure recordings, if available.
-
Reviewer 3 Report
Comments and Suggestions for Authors
The authors identified Slc26a6 as a molecule that regulates cardiac pH and found that ablation of Slc26a6 protects the heart from I/R injury. The study addresses an important topic with potential implications for understanding cardiac ion transport and pH regulation during I/R injury. The experimental design is generally good, and the data appears promising. However, several concerns remain, particularly inclusion of validation data confirming gene deletion, and consistency in sample size reporting are needed. The discussion should also focus more directly on the authors’ findings rather than providing general background. Finally, many references are outdated and should be replaced with more recent and relevant literature to better reflect current understanding.
Comments:
Results:
The authors are encouraged to include validation data (e.g., RT-qPCR or Western blot) demonstrating that Slc26a6 expression is effectively deleted or significantly reduced in cardiac tissue from Slc26a6⁻/⁻ mice used in this study. This information is essential to confirm the genetic manipulation and strengthen this study’s conclusions. These results could be incorporated into Figure 2 or presented as a Supplemental Figure.
Figure 2A:
Please add clear annotations to indicate which ECG trace (black or red) corresponds to WT and Slc26a6⁻/⁻ mice, respectively. In addition, an arrow should be included to highlight the ST-segment elevation within the trace for improved interpretability.
Figure 2B:
Please include labels specifying which curve (black or red) represents WT and Slc26a6⁻/⁻ mice to enhance figure clarity.
Figure 3B, D, E:
Please correct the figure legend so that the circles representing Slc26a6⁻/⁻ mice are green color and different from WT mice.
Figure 3:
The sample sizes between the WT I/R and Slc26a6⁻/⁻ I/R groups appear to be markedly different (5 vs. 9 in panel D, and 8 vs. 16 in panel E). Please clarify whether these unequal group sizes were accounted for in the statistical analysis and explain how this difference in animal number might affect the interpretation of the results. In addition, please clarify why several mice from the Slc26a6⁻/⁻ I/R group shown in Figure 3E (n = 16) appear to have been excluded from the analyses presented in Figures 3B and 3C, where only 5 and 9 mice were included, respectively. A brief justification for the exclusion criteria and consistency of sample numbers would improve the reliability of the data presentation.
Figure 4E:
Please revise the y-axis label to “LV Volume, S” instead of “Volume, S,” if this parameter represents left ventricular volume.
Figure 5:
It is recommended that the authors include representative traces from both WT and Slc26a6⁻/⁻ mice for left ventricular pressures (5A), left ventricular volumes (5B), and left ventricular pressure development (5C). Including these traces along with the quantitative graphs would allow readers to better see the physiological differences between the two groups.
Figure 6C, F:
Please clarify the statistical approach used for these analyses (a one-way ANOVA or a t-test was performed?). In addition, there appears to be no significant difference between the Sham and I/R groups for both WT and Slc26a6⁻/⁻ mice according to current one-way ANOVA statistical analysis. The authors should confirm this observation and provide a brief explanation or interpretation.
Figure 7A:
Please clarify the intent of this image. Is it intended to show the brightness of emission light? The figure should clearly indicate which visual features correspond to changes in pHi. It is recommended that the authors include comparative images of WT and Slc26a6⁻/⁻ mice at excitation wavelengths of 580 nm and/or 640 nm. This will help readers to see the differences in fluorescence or pHi more readily.
Figure 7B:
Please specify whether the pHi data presented were derived from cardiomyocytes of Sham or I/R mice. It is recommended that the authors include pHi measurements from both Sham and I/R conditions in WT and Slc26a6⁻/⁻ groups. This addition would facilitate a clearer comparison and provide stronger support for the conclusion that ablation of Slc26a6 is cardio-protection against I/R injury.
Discussion:
The discussion (particularly Sections 4.1 and 4.2) currently reads more like a general review of I/R injury rather than a focused discussion of the study’s findings. Please revise these sections to clearly connect the authors’ results to the respective topics.
References:
Approximately half of the cited references are outdated, with many published more than 15 years ago, including around 20 references from the 1990s (ref# 4-9, 22-23, 29-31, 43, 49-55, 58, 62). The authors should update the reference list by incorporating more recent and relevant studies from the past 10 years, if possible, to reflect current knowledge and strengthen the scientific context of the manuscript.
Round 2
Reviewer 1 Report
Comments and Suggestions for Authors
I have no further concerns.
Author Response
We thank the reviewer for the encouraging comments. Thank you very much.
Reviewer 3 Report
Comments and Suggestions for Authors
All responses are satisfactory except for the following issue: the authors did not provide validation data (e.g., RT-qPCR or Western blot) confirming that Slc26a6 expression is effectively deleted or reduced in the cardiac tissue of the Slc26a6⁻/⁻ mice used in this study. Without such data, it cannot be verified that the specific mouse cohort carries the intended genetic alteration or that the gene is truly inactivated in the heart.
While prior studies may serve as reference, new validation is generally required for each knockout study. Relying solely on data from eight years ago is not appropriate, as genetic drift over multiple generations (~5–6 years) can alter the genotype even within the same strain. The authors should therefore confirm that the current cohort exhibits the expected Slc26a6 loss in the heart tissue.
Round 3
Reviewer 3 Report
Comments and Suggestions for Authors
The authors have adequately addressed all of my previous concerns and questions.